# Memory-Enhanced Twin Delayed Deep Deterministic Policy Gradient (ME-TD3)-Based Unmanned Combat Aerial Vehicle Trajectory Planning for Avoiding Radar Detection Threats in Dynamic and Unknown Environments

**Jiantao Li** (ID)**, Tianxian Zhang** *(ID) **and Kai Liu** (ID)

School of Information and Communication Engineering, University of Electronic Science and Technology of China, Chengdu 611731, China; jiantaoli@std.uestc.edu.cn (J.L.); liukai19960417@msn.cn (K.L.)
* Correspondence: tianxianzhang@gmail.com or txzhang@uestc.edu.cn

**Abstract:** Unmanned combat aerial vehicle (UCAV) trajectory planning to avoid radar detection threats is a complicated optimization problem that has been widely studied. The rapid changes in Radar Cross Sections (RCSs), the unknown cruise trajectory of airborne radar, and the uncertain distribution of radars exacerbate the complexity of this problem. In this paper, we propose a novel UCAV trajectory planning method based on deep reinforcement learning (DRL) technology to overcome the adverse impacts caused by the dynamics and randomness of environments. A predictive control model is constructed to describe the dynamic characteristics of the UCAV trajectory planning problem in detail. To improve the UCAV's predictive ability, we propose a memory-enhanced twin delayed deep deterministic policy gradient (ME-TD3) algorithm that uses an attention mechanism to effectively extract environmental patterns from historical information. The simulation results show that the proposed method can successfully train UCAVs to carry out trajectory planning tasks in dynamic and unknown environments. Furthermore, the ME-TD3 algorithm outperforms other classical DRL algorithms in UCAV trajectory planning, exhibiting superior performance and adaptability.

**Keywords:** UCAV; trajectory planning; predictive control model; memory-enhanced twin delayed deep deterministic policy gradient

## 1. Introduction

In modern warfare, unmanned combat aerial vehicles (UCAVs) have gained increasing attention due to their advantages of high maneuverability, high stealth, and zero risk to pilots [1]. They are commonly deployed to execute hazardous missions such as aerial reconnaissance, penetration of defense systems, and target attacks [2,3]. Trajectory planning plays a crucial role in UCAV systems, where the UCAV constantly adjusts its flight state to generate an optimal route from the initial point towards the target area while avoiding threats from hostile aircraft, radars, and missiles [4]. UCAV trajectory planning is essentially a complicated nonlinear control problem with multiple constraints. Therefore, it is necessary to consider the balance between UCAV dynamic limitations, fuel consumption, and environmental threats to determine the best course of action.

In contrast to unmanned combat vehicle (UAV) trajectory planning for obstacle avoidance [5], UCAV trajectory planning for radar-detected threat avoidance entails several important features: (1) **Penetrability of the radar detection area**. Radar detection areas that the UCAV should avoid cannot be simply treated as impenetrable obstacles. In certain emergency situations, it may be justified for the UCAV to enter radar detection areas temporarily. This is because radar typically requires a certain amount of time to establish a tracking trajectory or guide a missile launch once a target is detected. (2) **Potential distribution of radar detection**

**threats**. Similar to the characteristics of the potential field, the probability of radar detection increases as the UCAV approaches radars [6]. Furthermore, if multiple radars overlap, a distinct calculation method is utilized based on the multi-radar fusion criteria [7]. (3) **Spatial configuration effect of the Radar Cross Section (RCS)**. When the UCAV and radar maintain a fixed distance and the radar operates with constant parameters, such as carrier frequency, transmit power, and pulse width, the probability of radar detection is determined solely by the RCS. This is influenced by the engagement geometry between the UCAV and the radar [8,9]. These characteristics exacerbate the complexity of UCAV control.

A considerable amount of research has been devoted to solving the UCAV trajectory planning problem. One of the most classical studies was carried out by Kabamba et al. [10]. Considering the coupling between the RCS and aircraft dynamics and the probabilistic nature of radar tracking, an optimal control model was established to describe UCAV trajectory planning. An efficient numerical optimization method was used to generate the optimal trajectory in the presence of radar and missile threats. The research method only provides discrete trajectories with a low accuracy. In [11], an improved intelligent water drop optimization algorithm was introduced to smooth the UCAV trajectory. These traditional numerical optimization algorithms have a low solving efficiency and cannot meet the requirements of real-time performance. To improve the accuracy of UCAV trajectory planning in a short period, more constructive and meaningful research has been conducted. Tang et al. proposed an online case-based trajectory planning strategy to achieve rapid solutions of control variables of a UCAV flight trajectory [12]. Furthermore, Wei et al. presented a novel approach to solve the formation of online collaborative trajectory planning for UCAVs using the HP adaptive pseudospectral method [13]. In [14], the complex UCAV trajectory planning problem was decomposed into a roadmap planning layer and an optimal control layer. In addition, a novel algorithm based on an updatable probabilistic roadmap (PRM) and a collision-free state-to-state trajectory planner based on the Gaussian pseudospectral method were proposed to solve these sub-layer problems, respectively. However, the above technologies have limited adaptability and are unable to meet higher real-time requirements in complex, uncertain, and dynamic battlefield environments. Fortunately, as an emerging type of machine learning, deep reinforcement learning (DRL) provides a viable solution to the UCAV trajectory planning problem.

DRL has been widely used to enhance the sensing and decision-making capabilities of unmanned systems because of its advantages in solving complex and high-dimensional problems. In the DRL framework, neural networks are employed as decision makers for agents, allowing them to take actions based on the current state and receive subsequent states and rewards from the environment. By continuously interacting with the environment, the agent updates the neural networks to learn the optimal policies [15]. In recent years, the widespread adoption of DRL technologies in engineering has propelled UCAVs towards acquiring autonomous capabilities in exploration, perception, and decision making, eliminating the need for human intervention [16]. Therefore, it is challenging to develop an intelligent and adaptive UCAV capable of dealing with complex, dynamic, and unknown battlefield environments.

In this paper, a DRL framework of UCAV trajectory planning for avoiding radar detection threats was constructed. We proposed a memory-enhanced twin delayed deep deterministic policy gradient (ME-TD3) algorithm to improve the UCAV's performance in dynamic and unknown environments. The main contributions and innovations of this paper are summarized as follows:

(1) UCAV trajectory planning for avoiding radar detection threats is modeled as a predictive control model. The model is described as a Partially Observable Markov Decision Process (POMDP) by constructing the state space, the observation space, the action space, and the reward function. The state space, the observation space, and the action space are designed to describe the UCAV's complete observations and partial observations of the environment and the UCAV's control vectors, respectively.

(2) We construct a sophisticated reward function to describe the impact of mission success, path loss, and radar detection threats. First, we normalize the reward to avoid gradient explosion of the neural network. Second, the path loss reward is designed in combination with the artificial potential field method to alleviate the sparse reward problem. Finally, to improve the learning efficiency of the agent in the early stage, the negative reward for crossing the flight boundary is removed.

(3) A memory-enhanced TD3 algorithm based on the attention mechanism is proposed to improve the performance of UCAV trajectory planning in the POMDP. In an unknown and dynamic environment, the information obtained by the UCAV is limited, which has a negative impact on the UCAV's decisions. By taking multi-step observations as input to the attention network, the UCAV can learn the dynamics of the environment in conjunction with historical information, which helps it to make better decisions.

The remainder of this paper is organized as follows: In Section 3, the system model is presented. In Section 4, we establish the predictive control model for UCAV trajectory planning. In Section 5, we introduce the memory-enhanced TD3 algorithm for UCAV dynamic trajectory planning. The simulation results are presented in Section 6. Further discussions are given in Section 7. The conclusions are given in Section 8.

## 2. Related Works

The application of DRL in UCAV dynamics, optimal control, and maneuver decisions is beginning to emerge. In recent studies, maneuver decision methods based on DRL algorithms were proposed to develop intelligent UCAVs for air combat [17,18]. A UCAV trained using DRL algorithms learned a series of basic maneuvers, such as diving, climbing, and circling, ultimately achieving a high winning rate against opponents. Focusing on the problem of the insufficient exploration ability of the DRL algorithm, Wang et al. proposed a UCAV air combat maneuver decision method based on a heuristic deep deterministic policy gradient (DDPG) algorithm [19]. In [20], Cao et al. studied autonomous maneuver decisions for UCAV air combat based on the double deep Q network algorithm (DDQN) and stochastic game theory, which further boosted the performance of the UCAV in different combat cases. To compensate for the low training efficiency caused by simple sampling mechanisms, Wang et al. proposed a task completion division soft actor–critic (TCD-SAC) algorithm for UAV penetration [21]. However, these studies did not take into account the uncertainty of environmental information obtained by agents in the real world, which leads to the degradation of DRL algorithm performance. To overcome the adverse impact of environmental fluctuations on decision making, Wan et al. proposed a Robust-DDPG to develop a robust UAV motion controller in dynamic, uncertain environments [22]. Li et al. proposed a meta twin delayed deep deterministic policy gradient (Meta-TD3) to realize the control of UAV maneuvering for target tracking and enable a UAV to quickly adapt to an uncertain environment [23]. Furthermore, in order to improve the cooperative capabilities of the UCAV swarm, multi-agent deep reinforcement learning (MADRL) algorithms were applied to large-scale air combat [24,25]. Based on centralized training with decentralized execution, MADRL provides a UCAV swarm with a high level of robustness.

Although DRL is widely used to address UAV trajectory planning problems, it presents novel challenges in complex combat environments, particularly those encompassing radar detection threats. Some researchers have paid attention to this problem and made preliminary attempts to solve it. In [26], a framework combining a DQN with prioritized experience replay (DQN-PER) and transfer learning was proposed to improve the performance of UAV path optimization under radar threats. In [27], to prove the importance of DRL in UAV trajectory planning, several classical DRL algorithms were implemented for a comparative analysis. In [28], the dual double deep Q-network (D3QN) algorithm was used for real-time UAV path planning. In a simulation environment based on the STAGE scenario software, the UAV showed excellent performance in both static and dynamic task settings.

These previous studies have achieved excellent results in UAV trajectory planning under the threat of radar detection, but they still have some drawbacks. Firstly, the impact of the UCAV's predictive ability on trajectory planning has not been thoroughly considered, which makes it difficult for a UCAV to make decisions further into the future when dealing with environmental dynamics and randomness. Secondly, the methods proposed in these studies simply make decisions based on the current state. Historical information has not been well utilized, resulting in poor performance and adaptability of trajectory planning. Thirdly, these studies have focused solely on the development of intelligent aircraft systems within known or single-scenario settings. The inherent potential of DRL for adaptability in dynamic and uncertain environments has not been well exploited. For the shortcomings above, we developed a predictive control model to describe the characteristics of UCAV trajectory planning for avoiding radar detection threats. A ME-TD3 algorithm is proposed to process the historical features of environmental information, effectively improving the UCAV's predictive ability. Then, we trained a UCAV in different mission scenarios, enabling it to adapt to dynamic and unknown battlefield environments.

## 3. System Model

In this section, we consider UCAV trajectory planning for avoiding radar detection threats in a penetration combat scenario shown in Figure 1. Specifically, the goal of the UCAV is to reach a designated target area for an attack mission. During this period, it must navigate through a hostile surveillance area that is actively monitored by ground-based radars (GBRs) and airborne radars (ARs). The probability of radar detection is correlated with the UCAV's attitude and position in relation to the radar. Therefore, it is imperative for the UCAV to continually adjust its flight attitude to safely traverse the surveillance area while evading radar detection threats. We assume that the UCAV can obtain real-time positions and parameters of the radars using its dedicated reconnaissance system. In the following subsections, we present the UCAV and radar motion model, the RCS model, and the radar detection threat model, which are used as the basic principle for the problem formulation in Section 4 and for DRL for UCAV trajectory planning in Section 5.

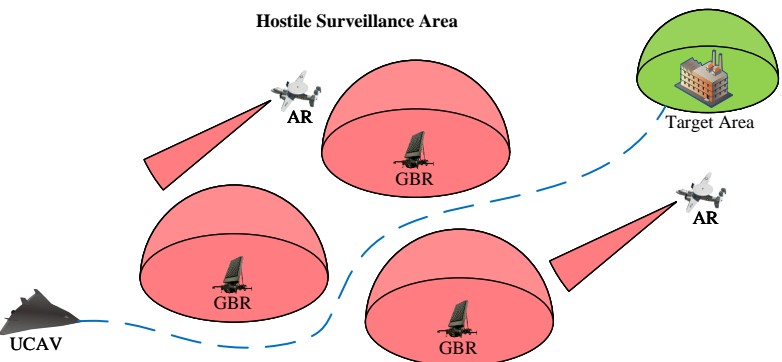

**Figure 1.** UCAV trajectory planning for avoiding radar detection threats in a penetration combat scenario.

### 3.1. UCAV and Radar Motion Model

The motion model of the UCAV exhibits a high degree of complexity. In this subsection, we consider a fixed-wing aircraft motion model, which eliminates the complex aerodynamic forces acting on the airframe [29]. We assume that the UCAV flies in a windless environment at a constant altitude. A stable correlation is established between the UCAV's coordinated turn and navigation direction, ensuring that the yaw angle and heading angle consistently maintain identical values.

The UCAV's position in the north–east–down (NED) coordinate frame $p_u^n$ and attitude $\Theta_u$ can be defined as

$$p_u^n = \begin{bmatrix} p_{un} & p_{ue} & p_{ud} \end{bmatrix}^\top \tag{1}$$

$$\mathbf{\Theta_u} = \begin{bmatrix} \phi_u & \theta_u & \psi_u \end{bmatrix}^\top \tag{2}$$

where $\mathbf{\Theta_u}$ is the vector of Euler angles for the roll, pitch, and yaw of the UCAV.

In the motion model, the pitch angle $\theta_u$ is constant. The motion equations of the UCAV are given by

$$\dot{p}_{un} = v_u \cos \psi_u \tag{3}$$

$$\dot{p}_{ue} = v_u \sin \psi_u \tag{4}$$

$$\dot{\psi}_u = \frac{g}{v_u} \tan \phi_u \tag{5}$$

where $g$ is the acceleration due to gravity and $v_u$ is the speed of the UCAV.

In the scenario of this paper, in order to cooperate with ground-based radars to monitor the protected airspace, airborne radars cruise along a certain path. Since the motion characteristics of the radars are not the focus of this study, we use a simple motion model to represent them. The position of the radar in the NED frame is given by

$$\mathbf{p_r^n} = \begin{bmatrix} p_{rn} & p_{re} & p_{rd} \end{bmatrix}^\top \tag{6}$$

The motion equation of the radar is given by

$$\dot{\mathbf{p}}_\mathbf{r}^\mathbf{n} = \mathbf{v_r} \tag{7}$$

where $\mathbf{v_r} = [v_{rn}\ v_{re}\ v_{rd}]^\top$ is the vector of radar speeds. When $\mathbf{v_r} = 0$, the type of radar is ground-based radar.

### 3.2. Radar Cross Section Model

The RCS is a non-linear function of the engagement geometry between the radar and the aircraft [30]. In this subsection, we consider a 3D ellipsoid RCS model [31], which gives an expression for the RCS as a function of the azimuth and elevation angles in the UCAV's body frame. The body frame's $x$, $y$, and $z$ axes extend forward, sideways, and downward from the center of the UCAV, respectively. We define the position of the radar in the body frame as

$$\mathbf{p_r^b} = \begin{bmatrix} p_{rx} & p_{ry} & p_{rz} \end{bmatrix}^\top. \tag{8}$$

The vector $\mathbf{p_r^b}$ is calculated using orientation according to the attitude of the UCAV by

$$\mathbf{p_r^b} = C_n^b(\mathbf{p_r^n} - \mathbf{p_u^n}) \tag{9}$$

where $C_n^b$ is the orthonormal rotation matrix from the NED frame to the body frame given by

$$C_n^b = \begin{bmatrix} C\psi_u C\theta_u & -C\phi_u S\psi_u + C\psi_u S\phi_u S\theta_u & S\phi_u S\psi_u + C\theta_u C\psi_u S\theta_u \\ C\theta_u S\psi_u & C\phi_u C\psi_u + S\phi_u S\psi_u S\theta_u & C\psi_u S\phi_u + C\phi_u S\psi_u S\theta_u \\ -S\theta_u & C\theta_u S\phi_u & C\phi_u C\theta_u \end{bmatrix} \tag{10}$$

where S· and C· represent the $\sin(\cdot)$ and $\cos(\cdot)$ functions, respectively.

The RCS azimuth angle $\lambda_r$ is defined as the angle between the body frame's $x$ axis and the direction of the radar transmit wave in the $x$–$y$ plane of the body frame. The RCS elevation angle $\phi_r$ is the angle between the $x$–$y$ plane in the body frame and the direction of the radar transmit wave. The RCS azimuth angle $\lambda_r$ and elevation angle $\phi_r$ are given by

$$\lambda_r = \arctan\left(\frac{p_{ry}}{p_{rx}}\right) \tag{11}$$

$$\phi_r = \arctan\left(\frac{p_{rz}}{\sqrt{(p_{rx})^2 - (p_{ry})^2}}\right). \tag{12}$$

The equation of a 3D ellipsoid RCS is given by

$$\sigma_r = \frac{abc}{\sqrt{(aS\lambda_r S\phi_r)^2 + (bC\lambda_r S\phi_r)^2 + (cC\phi_r)^2}} \tag{13}$$

where $\sigma_r$ is the RCS, and $a$, $b$, $c$ are the radii of the ellipsoid along the $x$, $y$, $z$ axes, respectively. Polar plots of the 3D ellipsoid RCS with respect to $\lambda_r$ and $\phi_r$ are shown in Figure 2.

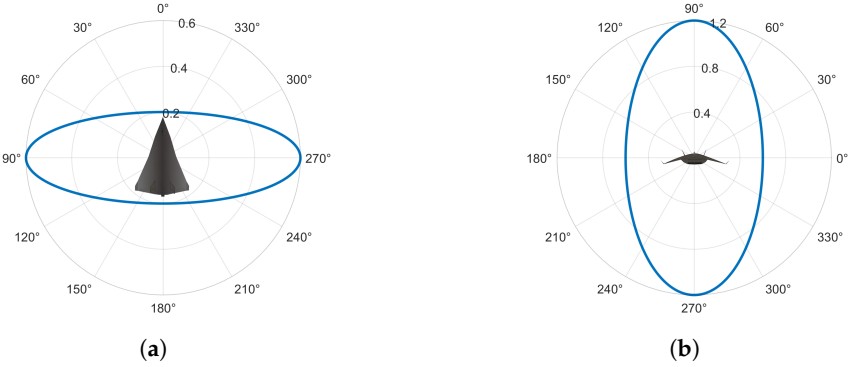

(**a**)  (**b**)

**Figure 2.** An example of a 3D ellipsoid RCS model. (**a**) RCS as function of azimuth angle $\lambda_r$. (**b**) RCS as function of pitch angle $\phi_r$.

### 3.3. Radar Detection Threat Model

The generation of radar detection threats comes from the ability to guide fighter aircraft and missiles against targets. Once the UCAV enters the radar detection range, it becomes vulnerable to detection and potential attacks by hostile air defense systems [32]. In this subsection, we establish a radar detection threat model based on the probability of radar detection. We assume that the UCAV has knowledge of the radar's transmit power, carrier frequency, antenna gain, etc.

For a single-pulse radar, the probability of radar detection $P_d$ is a function of the signal-to-noise ratio (SNR) and the probability of false alarms $P_{fa}$ [33]. Based on Marcum's Q-function, an approximate expression for $P_d$ is given by

$$P_d \approx 0.5 \times \text{erfc}\left(\sqrt{-\ln P_{fa}} - \sqrt{\text{SNR} + 0.5}\right) \tag{14}$$

where erfc($\cdot$) is the complementary error function given by

$$\text{erfc}(z) = 1 - \frac{2}{\sqrt{\pi}} \int_0^z e^{-\xi^2} d\xi. \tag{15}$$

An example curve of $P_d$ with respect to the SNR is shown in Figure 3. The SNR is given by

$$\text{SNR} = \frac{P_r}{P_n} \tag{16}$$

where $P_r$ is the radar receiver power and $P_n$ is the radar noise power.

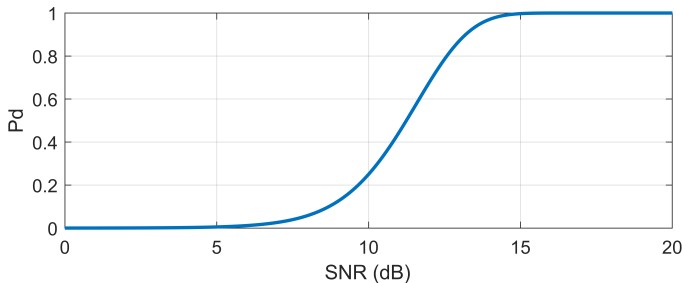

**Figure 3.** $P_d$ with respect to the SNR for a constant $P_{fa} = 10^{-6}$.

According to the radar equation, the radar receiver power $P_r$ is given by

$$P_r = \frac{P_t G_t^2 \lambda_c^2 \sigma_r}{(4\pi)^3 d^4} \tag{17}$$

where $P_t$ is the radar transmit power, $G_t$ is the radar antenna gain, $\lambda_c$ is the wavelength of the radar signal, and $d$ is the distance between the UCAV and the radar given by

$$d = \|\boldsymbol{p_u^n} - \boldsymbol{p_r^n}\|_2 \tag{18}$$

The radar noise power $P_n$ is given by

$$P_n = kT_0 B_n F_n \tag{19}$$

where $k$ is the Boltzmann constant $(1.38 \times 10^{-23} \text{J/K})$, $T_0$ is the noise temperature of the radar system, $B_n$ is the bandwidth of the radar receiver, and $F_n$ is the noise factor of the radar receiver. Expressing Equation (16) in terms of (17) and (19), we obtain

$$\text{SNR} = \frac{P_t G_t^2 \lambda_c^2 \sigma_r}{(4\pi)^3 d^4 kT_0 B_n F_n} = c_r \frac{\sigma_r}{kd^4}. \tag{20}$$

where $c_r$ is a constant determined by the radar parameters.

As the distance between the UCAV and the radar increases, the SNR decreases. When the SNR falls below a certain threshold, the radar is no longer able to effectively detect the UCAV. The maximum radar detection range is a term used to define the boundary of radar detection performance given by

$$d_{\max} = \left[ \frac{P_t G_t^2 \lambda_c^2 \sigma_r}{(4\pi)^3 S_{i\min}} \right]^{\frac{1}{4}} \tag{21}$$

where $S_{i\min}$ is the sensitivity of the radar receiver.

When the UCAV enters an area simultaneously covered by multiple radars, it is essential to consider the threat posed by collective radar detections. Using network radar plot fusion technology, the detection probability of the radar network is given by [34]

$$P_D = 1 - \prod_{i=1}^{N_r} \left( 1 - P_d^i \right) \tag{22}$$

where $P_d^i$ is the probability of $i$th radar detection and $N_r$ is the number of radars in the radar network. The equation of $P_D$ is a normal form expression. When $N_r = 1$, it is equal to the probability of single radar detection $P_d$.

It is evident that as the UCAV remains in the radar detection range and comes closer to the radar, the radar detection threat escalates. Therefore, the UCAV should try to avoid

entering the radar detection range as much as possible. The definition of the radar detection threat function is given by:

$$J = \int_{t_b}^{t_b+\tau} P_D(t)\,dt \tag{23}$$

where $t_b$ is the time at which the UCAV enters the radar detection range and $\tau$ is the period of time that the UCAV is present in the radar detection range. Once the radar detection threat exceeds a critical threshold (i.e., $J > J_c$), the UCAV is vulnerable to destruction by enemy air defense systems.

## 4. Problem Formulation

UCAV trajectory planning for avoiding radar detection threats in dynamic environments can be viewed as an optimal control problem (OCP). In this section, considering the intertwined influence of the UCAV's attitude on both the trajectory planning and the probability of radar detection and the variability of airborne radar positions, we present a predictive control model to accurately describe this OCP. Then, we characterize this OCP as a POMDP and design an appropriate state space, observation space, action space, and reward function.

### 4.1. Predictive Control Modeling for UCAV Trajectory Planning
#### 4.1.1. Optimal Control Modeling

In the UCAV trajectory planning problem, the system state $x$ consists of UCAV state $x_u$ and a radar state $x_r$, defined as

$$x = \begin{bmatrix} x_u & x_r \end{bmatrix}^\top,\ x(t_0) = x_0 \tag{24}$$

where $x_u = \begin{bmatrix} p_u^n & \Theta_u \end{bmatrix}^\top$ is the vector of the UCAV's position and attitude, and $x_r = \begin{bmatrix} p_r^n & v_r & c_r \end{bmatrix}^\top$ is the vector of the radar's position, speed, and parameters. $x_0$ is the initial value of the system state. The input control $u$ is defined as the roll angular velocity $\dot{\phi}_u$. The system dynamics are defined by the UCAV motion model and radar motion model in the previous section.

The control constraint is defined as

$$|\dot{\phi}_u| \le \dot{\phi}_{u\max} \tag{25}$$

where $\dot{\phi}_{u\max}$ is the maximum roll angular velocity.

The path constraint is defined by the boundary limits of the area and the UCAV's attitude, given by

$$x_{\min} \le x \le x_{\max} \tag{26}$$

where $x_{\min}$, $x_{\max}$ are the boundary of the system state.

The terminal constraint is defined by the boundary limit of the terminal region, given by

$$\left\| p_u^n(t_f) - p_f^n \right\|_2 \le R_f \tag{27}$$

where $p_u^n(t_f)$ is the UCAV's position at the final time $t_f$, and $p_f^n$ is the central position of the target area. The terminal constraint requires the UCAV to reach the target area of a circle with radius $R_f$.

The terminal cost is a negative constant for terminal reward $-R_f(x(t_f), t_f)$. The running cost is a function defined by the path loss and radar detection threats as the following:

$$\ell(x(t), u(t), t) = c_1 \left\| p_u^n(t) - p_f^n \right\|_2 + c_2 P_D(x(t), t) \tag{28}$$

where $c_1$ and $c_2$ are the weight coefficients of the running cost.

The optimal control problem for UCAV trajectory planning can be formulated as:

**Problem P1:**

$$\min_{\dot{\phi}_u(t)} \int_{t_0}^{t_f} c_1 \left\| \boldsymbol{p_u^n}(t) - \boldsymbol{p_f^n} \right\|_2 + c_2 P_D(\boldsymbol{x}(t), t) \, dt - R_f(\boldsymbol{x}(t_f), t_f)$$

$$\text{s.t.} \quad \begin{cases} \dot{p}_{un} = V_u \cos \psi_u \\ \dot{p}_{ue} = V_u \sin \psi_u \\ \dot{\psi}_u = \frac{g}{V_u} \tan \phi_u \\ \dot{\boldsymbol{p}}_r^n = \boldsymbol{v}_r \\ |\dot{\phi}_u| \le \dot{\phi}_{u\max} \\ \boldsymbol{x}_{\min} \le \boldsymbol{x} \le \boldsymbol{x}_{\max} \\ \left\| \boldsymbol{p_u^n}(t_f) - \boldsymbol{p_f^n} \right\|_2 \le R_f \end{cases} \tag{29}$$

### 4.1.2. Predictive Control Modeling

Typically, the decision controller in the optimal control model solely focuses on the current state of the system. However, there are some dynamic factors of environments that pose challenges to UCAV trajectory planning control as follows: (1) There is a coupled relationship between the UCAV's attitude $\boldsymbol{\Theta_u}$ and the probability of radar detection $P_D$. The UCAV dynamically adjusts its attitude to generate a trajectory toward the target area while avoiding radar detection threats. However, these adjustments simultaneously alter the $P_D$ due to changes in the RCS according to (14) and (20), necessitating the UCAV to readjust its attitude to generate a new trajectory accordingly. (2) The position of the airborne radar dynamically changes within the area that the UCAV must traverse. Under certain special conditions, the UCAV must take into account the movement of the airborne radar to devise its trajectory for a better future cost.

As shown in Figure 4a, in the first case, the UCAV developed based on a non-predictive control model cannot predict the impact of future attitudes on the radar detection range. It only makes temporary changes to environmental states. The UCAV developed based on a predictive control model has a global planning capability, allowing it to anticipate changes in the radar detection range and generate a shorter trajectory. Therefore, it is imperative to anticipate the impact of current actions on future states to improve the performance of UCAV trajectory planning.

As shown in Figure 4b, in the second case, we ignore the changes in the radar detection range. At time $t_1$, the UCAV developed based on a non-predictive control model tends to fly along a straight line toward the target area because it only focuses on the current position of the airborne radar. However, as the radar position changes, the UCAV has to adjust its expected trajectory to avoid the radar detection threat, resulting in a longer task time. The UCAV developed based on a predictive control model considers the future position of airborne radar and selects a shorter and smoother trajectory.

Failure to account for these critical factors will result in a degraded trajectory planning performance. Therefore, it is imperative for a UCAV to accurately predict future states and costs in order to develop a UCAV trajectory planning method with efficiency and safety. The basic principle of predictive control models is to use process models to predict the future state of the system under certain control effects. Based on this, the optimal control quantity is solved iteratively according to the given constraints and performance requirements. At each step of the iteration, real-time states are detected to correct predictions of future states. The expression of the state transition equation of the UCAV control system can be written as

$$\boldsymbol{x}(t+1) = \boldsymbol{f}(\boldsymbol{x}(t), \dot{\phi}_u(t), t) \tag{30}$$

where $\boldsymbol{f}$ is the state transition function of the UCAV control system. Unlike the previous optimal control model, the goal of the predictive control model is to minimize the future total costs by optimizing control inputs over a period of time. The predictive control model for UCAV trajectory planning is formulated as

**Problem P2:**

$$\min_{\dot{\phi}_u(t)} \sum_{k=1}^{N} \boldsymbol{F}(\boldsymbol{x}(t+k), \dot{\phi}_u(t+k), t+k)$$

$$\text{s.t.} \quad \dot{\boldsymbol{x}} = \boldsymbol{f}(\boldsymbol{x}(t), \dot{\phi}_u(t), t)$$

$$\boldsymbol{H}(\boldsymbol{x}(t), \dot{\phi}_u(t), t) \leqslant 0$$

(31)

where $\boldsymbol{F}$ and $\boldsymbol{H}$ are the system cost functions and the system constraint functions given in Problem **P1**, respectively.

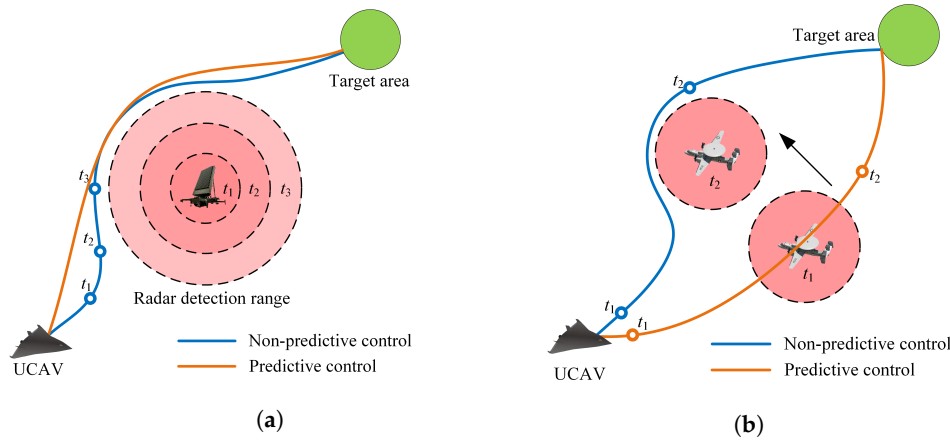

**(a)**                    **(b)**

**Figure 4.** Dynamic factors of the UCAV trajectory planning environment. (**a**) Dynamic variation in maximum radar detection range $d_{\max}$ with UCAV attitude. (**b**) Dynamic variation in the position of airborne radars.

### 4.2. POMDP Modeling for UCAV Trajectory Planning

In deep reinforcement learning, the agent learns the optimal policy using the experience gained through interaction with the environment. When the agent lacks knowledge of the environment and is unable to directly observe the complete state of the environment, the interaction process can be referred to as a POMDP [35]. A POMDP can be characterized by the following key elements:

- State space $\mathcal{S}$. Let $s_t \in \mathcal{S}$ represent the state at time $t$;
- Observation space $\mathcal{O}$. Let $o_t \in \mathcal{O}$ represent the observation of the agent at time $t$;
- Action space $\mathcal{A}$. Let $a_t \in \mathcal{A}$ represent the action of the agent at time $t$;
- State transition probability

$$p(s'|s, a) = p(s_{t+1} = s'|s_t = s, a_t = a)$$

(32)

  where $s, s' \in \mathcal{S}$ represent the current state and the next state, respectively, and $a \in \mathcal{A}$ represents the current action;
- Conditional observation probability

$$p(o|s, a) = p(o_t = o|s_t = s, a_t = a)$$

(33)

  where $o \in \mathcal{O}$ represents the current observation;
- Reward function

$$r(s, a) = r(s_t = s, a_t = a)$$

(34)

- Discount factor $\gamma \in [0, 1]$.

In a POMDP, the agent receives an observation $o$ correlated with the environment state $s$. Based on a stochastic initial policy $a \sim \pi(\cdot|o)$, the agent infers the state $s$ based on the observation $o$ to perform an action $a$. In response, it transits to the state $s$ with the state transition probability $p(s'|s, a)$ and the environment feeds back a reward $r(s, a)$. In particular, the observation sequence no longer satisfies the Markov property $p(o_{t+1}|o_t, a_t, o_{t-1}, a_{t-1}, \ldots, o_0, a_0) \neq p(o_{t+1}|o_t, a_t)$. We construct a POMDP for UCAV

trajectory planning by detailing the state space, observation space, action space, and reward function.

### 4.2.1. State and Observation Space

As described in Section 4, we design the states based on the UCAV and radar motion model. The state space is defined as:

$$\mathcal{S} = \{ p_u^n, \Theta_u; P_r^n, V_r, C_r \} \tag{35}$$

where $P_r^n = [p_{r1}^n, p_{r2}^n, \ldots, p_{rN}^n]^\top$ is the set of all radar position vectors, $V_r = [v_{r1}, v_{r2}, \ldots, v_{rN}]^\top$ is the set of all radar speed vectors, and $C_r = [c_{r1}, c_{r2}, \ldots, c_{rN}]^\top$ is the set of all radar parameter vectors.

The UCAV's position and attitude can be measured using an inertial navigation system, and real-time measurements of radar position, speed, and parameters can be obtained using a reconnaissance system. However, the cruise intention of airborne radars is difficult to predict. The observation space is defined as:

$$\mathcal{O} = \left\{ \overline{p}_u^n, \overline{\Theta}_u; \overline{P}_r^n, \overline{C}_r \right\} \tag{36}$$

where $\overline{p}_u^n, \overline{\Theta}_u$ are the measured values of UCAV state, $\overline{P}_r^n$ is the set of measured values for all radar position vectors, and $\overline{C}_r$ is the set of measured values for all radar parameter vectors. In this paper, we assume that the measurement error values of the UCAV and radar states can be ignored.

### 4.2.2. Action Space

According to UCAV dynamics, the UCAV takes actions to control its flight attitude for trajectory planning. In this paper, the action space is continuous, meaning that it consists of a range of real-valued or continuous options, rather than being limited to a discrete set of choices. The action space is defined as:

$$\mathcal{A} = \dot{\phi}_u, \; |\dot{\phi}_u| \leq \dot{\phi}_{u\max} \tag{37}$$

Compared to the control vectors of real aircraft systems, the action space is simplified to implement control optimization of UCAV dynamic trajectory planning.

### 4.2.3. Reward Function

The reward function plays a crucial role in deep reinforcement learning (DRL) as it directly affects the agent's ability to learn the optimal policy. In this article, we designed a precise reward function to improve the optimization efficiency by drawing on the human experience of UCAV trajectory planning. The reward function consists of three sub-rewards: (1) The primary mission goal of UCAV is to reach the target area. When the UCAV reaches the target area, it should receive a higher reward. (2) Considering the limited mission time and fuel consumption, the UCAV should reach the target area as soon as possible. The more the UCAV deviates from the target area and the longer it flies, the more severe the punishment it faces. (3) The UCAV should try to avoid entering the radar detection range as much as possible. If it is already within the radar detection range, it will be subject to a continuous punishment until it flies out of the threat area. The reward function is designed as the weighted sum of three sub-reward functions as follows:

$$r(s, a) = \omega_1 r_{\text{final}}(s, a) + \omega_2 r_{\text{path}}(s, a) + \omega_3 r_{\text{threat}}(s, a) \tag{38}$$

where $r_{\text{final}}$ is the final reward function for reaching the target area, $r_{\text{path}}$ is the negative reward function for path loss based on the distance between the UCAV and the target area, and $r_{\text{threat}}$ is the negative reward function for radar detection threats, which will keep increasing as the radar exposure time increases. $\omega_1, \omega_2, \omega_3$ are the weights of the sub-reward functions, respectively,

and satisfy $\sum \omega_i = 1$. The balance between different sub-reward functions is important. If the weight assigned to the final reward and path loss is higher, the UCAV will ignore the radar detection threat and take risks to cross the radar detection area to reach the target area faster. On the contrary, the UCAV will focus too much on flight safety. It will cautiously avoid radar detection areas that are far away, which will reduce the efficiency of mission execution.

The expression of the sub-reward functions is defined as:

$$
\begin{aligned}
r_{\text{final}}(s,a) &= \begin{cases} 1, & \text{if} \left\| \boldsymbol{p}_u^n(t_f) - \boldsymbol{p}_f^n \right\|_2 \leq R_f \\ 0, & \text{else} \end{cases} \\
r_{\text{path}}(s,a) &= - \left( \left\| \boldsymbol{p}_u^n - \boldsymbol{p}_f^n \right\|_2 \right)_{\text{norm}} \\
r_{\text{threat}}(s,a) &= \begin{cases} -(J)_{\text{norm}}, & \text{if } \forall d_i \leq d_{i\text{max}} \\ 0, & \text{else} \end{cases}
\end{aligned}
\tag{39}
$$

where $d_i$ is the distance between the UCAV and the $i$th radar, and $d_{i\text{max}}$ is the maximum effective detection range of the $i$th radar. $(\cdot)_{\text{norm}}$ represents normalization as follows:

$$
(x)_{\text{norm}} = \frac{x - x_{\text{min}}}{x_{\text{max}} - x_{\text{min}}}
\tag{40}
$$

where $x_{\text{min}}, x_{\text{max}}$ are the minimum and maximum value of $x$, respectively. The function converts the original value to the range $[0,1]$ linearly.

**Remark 1.** *The three sub-reward functions are constrained to the same magnitude. We set the value of the reward for reaching the target area as 1 and normalize the rewards for path loss and radar detection threats, which will avoid gradient explosion of the neural network.*

**Remark 2.** *The combination of the reward for path loss and the artificial potential field method is proposed. In the majority of maze games, in order to support the agent in reaching the target area as soon as possible, the path loss payoff is usually set to a constant such as $-1$. The disadvantage is that when the number of steps from the start to the target area is particularly large, it is difficult for the agent to explore the target area (i.e., sample a positive reward). The result is that the agent poorly learns to find the trajectory to the target area. To alleviate the sparse reward problem, we use a negative distance between the UCAV and the target area to design a reward for path loss, which seeks to provide effective and persistent proximity and direction feedback for reaching the target area right from the start of each episode.*

**Remark 3.** *The negative reward for crossing the flight boundary is removed. The boundary penalty is a typical method of limiting the agent's futile exploration, which encourages the agent to learn the optimal trajectory in an effective region. However, in the early stages of DRL training, the agent spends so much time moving away from the flight boundary rather than finding the optimal trajectory. Fortunately, with the guidance of the path loss reward based on an artificial potential field, the UCAV is able to find the optimal trajectory to the target area more quickly, even without the boundary constraint.*

## 5. Deep Reinforcement Learning for UCAV Trajectory Planning

Our problem is that the UCAV adjusts its attitude to reach the target area as quickly as possible while avoiding radar detection threats. However, it is not easy to address this problem due to the uncertainty and variability of dynamic environments. Fortunately, deep reinforcement learning (DRL) provides an effective solution for the sequential decision-making problem with an unknown and partially observable environmental model. In this section, a memory-enhanced twin delay deep deterministic policy gradient (ME-TD3) algorithm is proposed to generate UCAV motion control policies with predictive and generalization abilities.

*5.1. Overview of Deep Reinforcement Learning and the Actor–Critic Framework*

5.1.1. Deep Reinforcement Learning

Deep reinforcement learning is a recognized solution for the POMDP problem. The goal of DRL is to find an optimal policy that maximizes the cumulative return so as to maximize the long-term benefits in the decision-making process. The return is described by the discounted return at time *t*, given by

$$R_t = \sum_{k=0}^{\infty} \gamma^k r_{t+k} \tag{41}$$

The expectation of the discounted return is employed to estimate the policy value. The action–value function is defined as the expected return of taking action *a* according to the policy $\pi$ in a specific state *s* [36], as follows:

$$Q^{\pi}(s, a) = \mathbb{E}_{\pi}[R_t | s_t = s, a_t = a] \tag{42}$$

The Bellman equation of the action–value function is defined as:

$$Q^{\pi}(s, a) = r + \gamma \mathbb{E}_{\pi}[Q_{\pi}(s', a')] \tag{43}$$

The agent earns the highest expected return by exploring actions and updating policies. Among all possible policies, the optimal policy $\pi^*$ is the one that maximizes the action–value function. It is expressed as follows:

$$\pi^* = \arg \max_{\pi} Q_{\pi}(s, a) \tag{44}$$

The optimal action–value function is defined as:

$$Q^*(s, a) = r + \gamma \mathbb{E}_{\pi}[\max_{a' \in \mathcal{A}} Q^*(s', a')] \tag{45}$$

5.1.2. Actor–Critic Frameworks

Actor–critic is a classical reinforcement learning framework that combines a policy gradient (actor) and a value function (critic) to optimize decision making in continuous action spaces [37]. The actor learns the optimal policy, while the critic evaluates its performance and provides guidance for improvement. In the actor–critic framework, the policy network $\pi_{\phi}$ can be updated through the deterministic policy gradient algorithm:

$$\nabla_{\phi} J(\phi) = \mathbb{E}_{s \sim p_{\pi}} \left[ \nabla_a Q^{\pi}(s, a)|_{a=\pi(s)} \nabla_{\phi} \pi_{\phi}(s) \right] \tag{46}$$

The action value is estimated by a value network approximator $Q_{\theta}(s, a)$, with parameters $\theta$. In deep deterministic policy gradient (DDPG) [38], the value network is updated using temporal difference learning with a target network $Q_{\theta'}(s, a)$ to minimize the loss:

$$L(\theta) = \mathbb{E}_a \left[ (y - Q_{\theta}(s, a))^2 \right] \tag{47}$$

where

$$y = r + \gamma Q_{\theta'}(s', a'), \quad a' \sim \pi_{\phi'}(s') \tag{48}$$

and where the action $a'$ is taken according to a target actor network $\pi_{\phi'}$. The weights of the target networks are updated by a certain proportion at each time step:

$$\phi' \leftarrow \tau \phi + (1 - \tau)\phi' \tag{49}$$

$$\theta' \leftarrow \tau \theta + (1 - \tau)\theta' \tag{50}$$

where $\tau$ is the soft update factor.

*5.2. Twin Delay Deep Deterministic Policy Gradient*

Twin delay deep deterministic policy gradient (TD3) improves deep deterministic policy gradient (DDPG) by adding three training tricks as follows [39]:

*(1)    Clipped Double Q-Learning*

Due to the presence of noise, the Q value estimation in DDPG is prone to overestimation. This overestimation poses a risk, as the accumulated estimate errors can lead an agent converging to a local optimum or suffer from catastrophic forgetting during training. Consequently, DDPG may become more unstable when confronted with challenging tasks. To tackle the problem of overestimation bias, TD3 proposes a clipped double Q-learning approach. Within the actor–critic framework, a target policy network and a pair of target value networks are used to update the action value, shown as:

$$y_1 = r + \gamma Q_{\theta'_1}(s', \pi_{\phi'}(s')) \tag{51}$$

$$y_2 = r + \gamma Q_{\theta'_2}(s', \pi_{\phi'}(s')) \tag{52}$$

where $\theta'_1$, $\theta'_2$ are the parameters of the target value networks.

Then, the smaller of the two estimations is chosen as the target value update:

$$y = r + \gamma \min_{i=1,2} Q_{\theta'_i}(s', \pi_{\phi'}(s')) \tag{53}$$

Although choosing a lower Q value during network updating might introduce a potential underestimation bias, this bias is not explicitly propagated through the policy update. Therefore, the error is significantly reduced compared to the original training method.

*(2)    Target Networks and Delayed Policy Updates*

The target networks serve as deep function approximators, contributing to the algorithm's stability. Deep function approximators typically require several gradient updates to converge, while target networks offer a steady target during the updating process and enable the network to adapt to a broader range of training data.

To ensure effective training, the policy network is updated at a slower rate compared to the value network. This delay in policy updates ensures that the policy is not modified until the value error is minimized through updates from the value network. Additionally, to maintain a small error, the policy and target networks are updated after a fixed number of value network updates. By appropriately delaying the policy updates, we minimize the likelihood of repeatedly updating an unchanged policy. This approach reduces the variance in the value estimate, resulting in higher-quality policy updates. In essence, less frequent policy updates yield value estimates with lower variance, thereby enhancing the overall quality of the policy updates.

*(3)    Target Policy Smoothing Regularization*

When overfitting to narrow peaks in the value estimate, a deterministic target policy can be vulnerable to inaccuracies caused by function approximation errors, which increases the variance in the target policy. To reduce the variance, a regularisation strategy is introduced to smooth the target policy. In practice, a small amount of random noise is added to the target policy. The target value update is modified as follows:

$$y = r + \gamma \min_{i=1,2} Q_{\theta'_i}(s', \pi_{\phi'}(s') + \epsilon)$$
$$\epsilon \sim \text{clip}(\mathcal{N}(0, \sigma), -l, l) \tag{54}$$

where $\epsilon$ is normal distributed noise with zero mean and $\sigma$ variance. The added noise is clipped at $[-l, l]$ to keep the target action close to the original action.

### 5.3. Memory-Enhanced Twin Delay Deep Deterministic Policy Gradient

The dynamics of the UCAV and radar states in the process of UCAV trajectory planning introduce serious uncertainty and complexity into UCAV motion control. Therefore, it is essential to integrate the historical trajectory for the agent to learn an optimal policy. From the previous predictive control model, we know that the radar detection range is determined by its operating parameters and the UCAV's RCS. The UCAV's RCS can be calculated by its position relative to the radar and its attitude. When the UCAV takes action to change its position and attitude, the radar detection range also changes accordingly. This makes it necessary for the UCAV to predict the impact of current observations and actions on future states, planning a flight trajectory in advance to reach the target area while effectively avoiding radar detection. In a single scenario, based on the expected return mechanism of reinforcement learning, it is easy for the UCAV to predict future environmental changes solely based on the current observations and actions. However, in random and unknown scenarios, the impact of current observations and actions on future states is unpredictable. Therefore, it is very important to utilize the features of historical observations and actions to obtain the patterns of radar detection range and position changes and improve the trajectory planning performance of UCAVs. As shown in Figure 5, we proposed a memory-enhanced twin delay deep deterministic policy gradient (ME-TD3) to process the historical features of observations and actions.

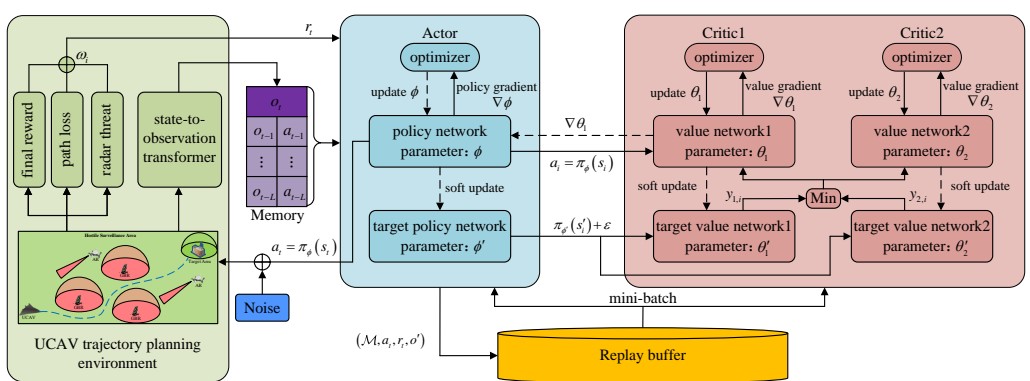

**Figure 5.** Illustration of the proposed algorithm.

On the basis of TD3, we construct a memory of size $L$ to collect historical observations and actions. At time $t$, the contents of memory $\mathcal{M}$ are generated through a sequence of observations $h_{o,t} = o_t, o_{t-1}, o_{t-2}, \ldots, o_{t-L}$ and actions $h_{a,t-1} = a_{t-1}, a_{t-2}, \ldots, a_{t-L}$. The memory is utilized as input for the actor network to update the current action $a_t$. Then, we collect the reward $r_t$ and the next observation $o_{t+1}$ obtained through interaction with the environment. During the interaction with the environment, the observations and actions obtained are first stored in the memory. When the amount of data in the memory exceeds the maximum capacity limitation, the bottom observation and action will be deleted and the data in the memory will be updated. The transition tuple $(\mathcal{M}, a, r, o')$ is stored in the replay buffer $\mathcal{B}$. The actor and critic networks utilize the experience randomly sampled from the replay buffer to learn optimal policies. The actor network takes action $a$ based on historical observations $h_o$ and then updates the next historical observations $h'_o$. The critic network updates the action value based on the next historical memory $\mathcal{M}'$ and noise action $\tilde{a}$, shown as:

$$
\begin{aligned}
y &= r + \gamma \min_{i=1,2} Q_{\theta'_i}(\mathcal{M}', \tilde{a}) \\
\tilde{a} &\leftarrow \pi_{\phi'}(h'_o) + \varepsilon, \varepsilon \sim \text{clip}(\mathcal{N}(0, \sigma), -l, l) \\
\theta_i &\leftarrow \text{argmin}_{\theta_i} N^{-1} \Sigma (y - Q_{\theta_i}(\mathcal{M}, a))^2
\end{aligned}
\tag{55}
$$

Similar to the TD3 algorithm, the actor network and target networks are updated after a *K*-step update of the critic network. The parameters of the actor and critic networks are updated, shown as:

$$\nabla_\phi J(\phi) = N^{-1} \Sigma \nabla_a Q_{\theta_1}(\mathcal{M}, a)|_{a=\pi_\phi(h_o)} \nabla_\phi \pi_\phi(h_o) \tag{56}$$

In this algorithm, we design actor and critic networks based on an attention mechanism, as shown in Figure 6. The historical observations and actions are normalized in the range $[-1, 1]$ as the inputs of the actor network and the critic network, respectively. In the actor network, the normalized observations are linearized by two linear network layers with 128 nodes. Then, the linearized vectors are processed by an attention network. Finally, the vectors are linearized by linear network layers with 128 nodes and 1 node, respectively. The first three linear layers are followed by a Rectified Linear Unit (ReLU) activation function, and the fourth linear layer is followed by a hyperbolic tangent (tanh) activation function to output a current action $a_t$ in the range $[-1, 1]$. There are two differences in the critic network that distinguish it from the actor network. First, after the normalized observations and actions are linearized by linear network layers, the linearized vectors are concatenated. Second, there is no activation function following the fourth linear layer.

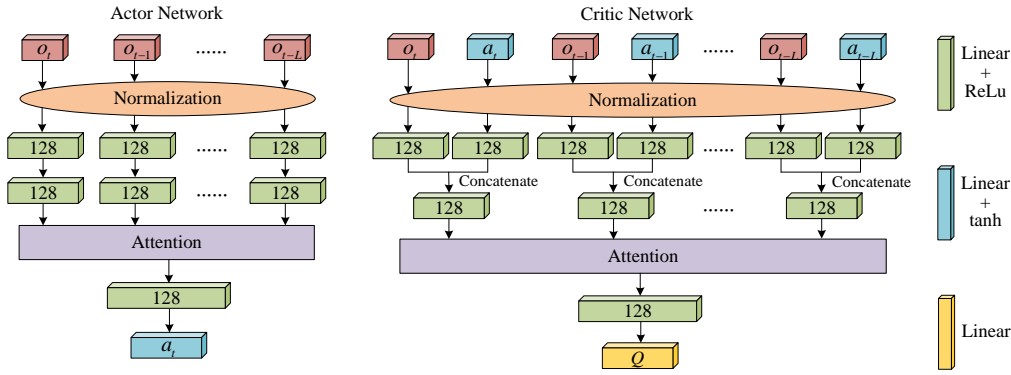

**Figure 6.** The network structure of the proposed algorithm.

In deep learning, attention mechanisms are usually applied to the processing of sequential data. The attention mechanism allows the model to assign different weights to different positions of the input sequence in order to focus on the most relevant part when processing each sequence element. We use an additive attention mechanism to obtain the context vector [40]. Suppose the input sequence is $M = (m_1, m_2, \ldots, m_n)$, where $m_i \in \mathbb{R}^d$ represents the *i*th state vector. The weight vector $e = (e_1, e_2, \ldots, e_n)$ can be computed by

$$e_i(m_i) = V_m^\top tanh(W_m \cdot m_i + b_m) \tag{57}$$

where $V_m, W_m, b_m$ are trainable parameters. The softmax function is used to normalize these weights:

$$\alpha_i = \frac{e_i}{\sum_{j=i}^n e_j}, \ \forall i = 1, 2, \ldots, n \tag{58}$$

Then, a context vector is computed as the output of the attention network layer as follows:

$$c_{att} = \sum_{i=1}^n \alpha_i \cdot m_i \tag{59}$$

The importance of appropriate weights for historical observations and actions can be understood through some examples. When a UCAV evades radar detection threats, the radar detection range changes with the UCAV's attitude. If the UCAV only focuses on the current observation, it cannot predict whether the radar detection range will increase or decrease in the future. By utilizing historical information, it can sensitively perceive

the changing trends in the radar detection range. Secondly, the cruise trajectory of the airborne radar is also unknown, and its future flight intentions can easily be predicted by processing the historical position of the airborne radar. The improved TD3 algorithm based on a memory-enhanced mechanism can better utilize the features of historical observations and actions, helping UCAVs better plan future control policies.

A summary of ME-TD3 for UCAV trajectory planning is given in Algorithm 1.

---

**Algorithm 1** ME-TD3 for UCAV trajectory planning

---

1: Randomly initialize actor network $\pi_\phi$, and critic networks $Q_{\theta_1}$, $Q_{\theta_2}$ with parameters $\phi$, $\theta_1$, $\theta_2$
2: Initialize target networks $\phi' \leftarrow \phi$, $\theta'_1 \leftarrow \theta_1$, $\theta'_2 \leftarrow \theta_2$
3: Initialize replay buffer $\mathcal{B}$, memory $\mathcal{M}$, batch size $N$, target update interval $K$
4: **for** episode = 1 to $T$ **do**
5: 　　Reset the environment
6: 　　**while** not done **do**
7: 　　　　Select action with exploration noise $a \sim \pi_\phi(h_o) + \varepsilon, \varepsilon \sim \mathcal{N}(0, \sigma)$ and observe reward $r$ and new state $o'$
8: 　　　　Store transition tuple $(\mathcal{M}, a, r, o')$ in $\mathcal{B}$ and $a, o'$ in $\mathcal{M}$
9: 　　**end while**
10: 　　Sample mini-batch of $N$ transitions $(\mathcal{M}, a, r, o')$ from $\mathcal{B}$
11: 　　$\tilde{a} \leftarrow \pi_{\phi'}(h'_o) + \varepsilon, \varepsilon \sim \text{clip}(\mathcal{N}(0, \tilde{\sigma}), -l, l)$
12: 　　$y \leftarrow r + \gamma \min_{i=1,2} Q_{\theta'_i}(\mathcal{M}', \pi_{\phi'}(h'_o))$
13: 　　Update critic networks $\theta_i \leftarrow \text{argmin}_{\theta_i} N^{-1} \Sigma (y - Q_{\theta_i}(\mathcal{M}, a))^2$
14: 　　**if** episode mod $K$ = 0 **then**
15: 　　　　Update $\phi$ by the deterministic policy gradient:
16: 　　　　$\nabla_\phi J(\phi) = N^{-1} \Sigma \nabla_a Q_{\theta_1}(\mathcal{M}, a)|_{a=\pi_\phi(h_o)} \nabla_\phi \pi_\phi(h_o)$
17: 　　　　Update target networks:
18: 　　　　$\phi' \leftarrow \tau \phi + (1 - \tau)\phi'$
19: 　　　　$\theta'_i \leftarrow \tau \theta_i + (1 - \tau)\theta'_i$
20: 　　**end if**
21: **end for**

---

## 6. Results

In this section, we present the situation environment used for UCAV trajectory planning and conduct targeted experiments to evaluate the proposed algorithm. First, the training results are shown to analyze the performance of the proposed algorithm. Second, the flight trajectory of the UCAV trained by the proposed algorithm is demonstrated to verify the algorithm's effectiveness. Third, the proposed algorithm is tested in different scenarios to verify its adaptability. All computations were executed on the same workstation with an AMD Ryzen threadripper 3970X CPU and an NVIDIA RTX A6000 GPU. The operating system is Ubuntu 20.04 and the computing architecture is CUDA 11.6. All of the experiments are performed under Python 3.9 and Pytorch 1.8.0. The visual display is based on the Matplotlib library.

### 6.1. Experimental Environment and Settings

To facilitate the training and testing of the DRL algorithm for UCAV trajectory planning, we developed a general simulation environment, shown in Figure 7. The mission space of the UCAV is a square area, subject to surveillance by hostile ground-based radars and airborne radars. Airborne radars fly back and forth along the *x*-axis or *y*-axis with constant speeds. The red areas represent the effective detection ranges of the radars. The green area represents the target area. The blue line represents the UCAV's flight trajectory. The black arrows represent the flight directions of airborne radars. The UCAV departs from a start point and navigates through a hostile surveillance area, eventually reaching the designated target area. It is obvious that the radar detection range alters as the UCAV's attitude changes. The

parameters of the UCAV motion model and the RCS model are shown in Table 1. We assume that the operating parameters of both the ground-based radars and the airborne radars are known and consistent within the same radar type, shown in Table 2. The structures of the actor and critic networks are shown in Figure 6. The parameters of radars and the UCAV are designed by referencing real-world data. The hyperparameters of the proposed algorithm are shown in Table 3.

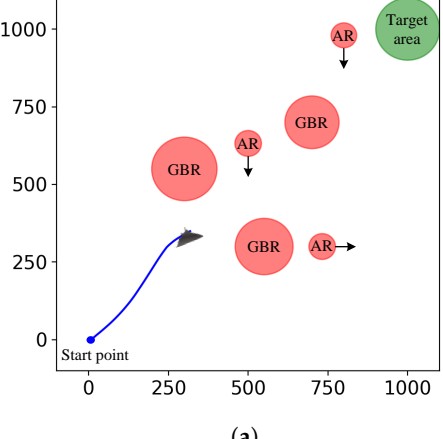
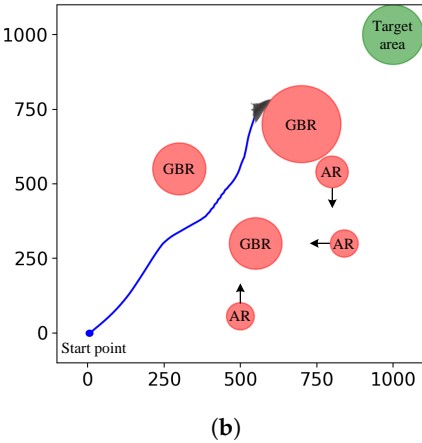

(**a**)                                                                                                  (**b**)

**Figure 7.** UCAV trajectory planning simulation environment. (**a**) The early stage of trajectory planning. (**b**) The later stage of trajectory planning.

**Table 1.** Parameters of the UCAV.

| Parameter | Value |
|---|---|
| Size of mission area | 1000 km $\times$ 1000 km |
| Time step of decision | 20 s |
| Flight altitude $h_f$ | 20 km |
| Max velocity $v_{u\max}$ | 0.5 km/s |
| Pitch $\theta_u$ | 0° |
| Range of yaw $\psi_u$ | $(-180°, 180°)$ |
| Range of roll $\phi_u$ | $(-90°, 90°)$ |
| Max roll velocity $\dot{\phi}_{u\max}$ | $\pi/3$ rad/s |
| Radii of the ellipsoid RCS $a, b, c$ | $(0.1, 0.5, 1)$ |

**Table 2.** Parameters of radars.

| Parameter | Value | |
|---|---|---|
| | **Ground-Based Radar** | **Airborne Radar** |
| Transmit power $P_r$ | $32 \times 10^6$ W | $30 \times 10^3$ W |
| Antenna gain $G_t$ | $10^{4.5}$ | $10^4$ |
| Wavelength $\lambda_c$ | 0.1 m | 0.1 m |
| Noise temperature $T_0$ | 290 K | 290 K |
| Bandwidth of receiver $B_n$ | 40 MHz | 5 MHz |
| Noise factor of receiver $F_n$ | $10^{0.5}$ | $10^{0.1}$ |
| False alarm probability $P_{fa}$ | $10^{-6}$ | $10^{-6}$ |

**Table 3.** Hyperparameters of the proposed algorithm.

| Hyper Parameter | Value |
|---|---|
| Max episodes | $500 \times 10^3$ |
| Max steps | 200 |
| Discount factor $\gamma$ | 0.99 |
| Soft update factor $\tau$ | 0.01 |
| Actor learning rate | 0.001 |
| Critic learning rate | 0.001 |
| Replay buffer size | $10 \times 10^6$ |
| Memory size | 3 |
| Batch size | 4096 |
| Max policy noise | 1 |
| Action bound | $\pi/6$ |
| Noise delay factor | 0.9999 |
| Delayed policy update interval | 3 |
| Target update interval | 100 |

### 6.2. Experiment of Algorithm Performance

In order to demonstrate the performance of the proposed algorithm, it is essential to perform appropriate comparative experiments. We use DDPG and TD3 as baseline comparisons, implementing them with almost the same hyperparameters as ME-TD3. To avoid the influence of random numbers, all agents for the three algorithms were trained in the same simulation environment with 10 random seeds. At the beginning of each episode, the starting positions of the UCAV and the radars are randomly initialized within the square area, and the flight speed of the airborne radar is randomly initialized within a range of (0.4 km/s, 0.6 km/s).

To evaluate the algorithms' performance, we used several quantitative evaluation metrics: the hit rate, crash rate, lost rate, and average reward. Specifically, the hit rate denotes the percentage of successful missions to the target area, the crash rate denotes the percentage of encounters with radar detection threats resulting in a crash, and the lost rate denotes the percentage of episodes where the vehicle remains trapped until the episode ends. These values were calculated over the most recent 500 episodes. Furthermore, we measured the average reward by calculating the mean value of the total rewards attained during the last 100 episodes. Due to the severe fluctuation in these evaluation indicators caused by random initialization and action noise, we processed the experimental results using the weight smoothing method with a weight factor of 0.9.

The convergence curves of the three algorithms are shown in Figure 8. It is evident that the agent explores the environment haphazardly in the initial training phase, leading to a remarkably low average reward. As the agent engages more frequently with the environment, it acquires more knowledge and improves its policy. The average reward curves of ME-TD3 tend to converge by approximately the 70,000th episode, whereas DDPG and TD3 typically converge by around the 100,000th episode. As training advances, the curves for ME-TD3 exhibit less fluctuation when compared to the other algorithms. The proposed algorithm demonstrates a faster convergence rate and a more stable convergence process in comparison to both DDPG and TD3. In the final training stage of the ME-TD3 algorithm, the hit rate is above 95%, the crash rate is below 3%, and the lost rate is less than 2%. ME-TD3 performs better compared to other algorithms in terms of the hit rate and crash rate. The loss rates of ME-TD3 and TD3 are generally the same, but both are significantly lower than that of DDPG. For further verification, we calculated the hit rates, crash rates, lost rates, and average rewards of different algorithms in the convergence phase. To avoid the impact of outliers on the performance analysis, we removed the maximum and minimum values of each indicator. The results of the best, worst, median, and mean values of these indicators under different random seeds are shown in Table 4.

Looking at the results, it is clear that ME-TD3 outperforms the other algorithms in terms of the hit rate, crash rate, and average reward. The best value of the lost rate of ME-TD3 is the smallest, and the worst, median, and mean values of TD3 are the smallest. The lost rate of ME-TD3 is close to that of TD3, but significantly lower than that of DDPG.

The reason for this is that the ME-TD3 effectively processes historical information using attention networks, which allows it to better adapt to dynamic and unknown environments.

**Table 4.** The overall results of the algorithms.

| Algorithm | DDPG [38] | | | | TD3 [39] | | | | ME-TD3 | | | |
|---|---|---|---|---|---|---|---|---|---|---|---|---|
| | HR | CR | LR | AR | HR | CR | LR | AR | HR | CR | LR | AR |
| Best | 92.52% | 4.410% | 2.780% | −39.34 | 96.21% | 3.570% | 0.140% | −35.32 | **97.06%.** | **2.720%** | **0.090%** | **−33.31** |
| Worst | 89.57% | 4.830% | 5.550% | −43.28 | 95.45% | 3.830% | **0.890%** | −37.14 | **95.95%** | **3.140%** | 0.930% | **−36.53** |
| Median | 90.90% | 4.620% | 4.800% | −42.34 | 95.96% | 3.720% | **0.320%** | −36.40 | **96.41%** | **2.900%** | 0.510% | **−35.05** |
| Mean | 90.79% | 4.630% | 4.520% | −41.89 | 95.93% | 3.700% | **0.410%** | −36.27 | **97.06%** | **2.920%** | 0.490% | **−35.08** |

The best data are presented in bold.

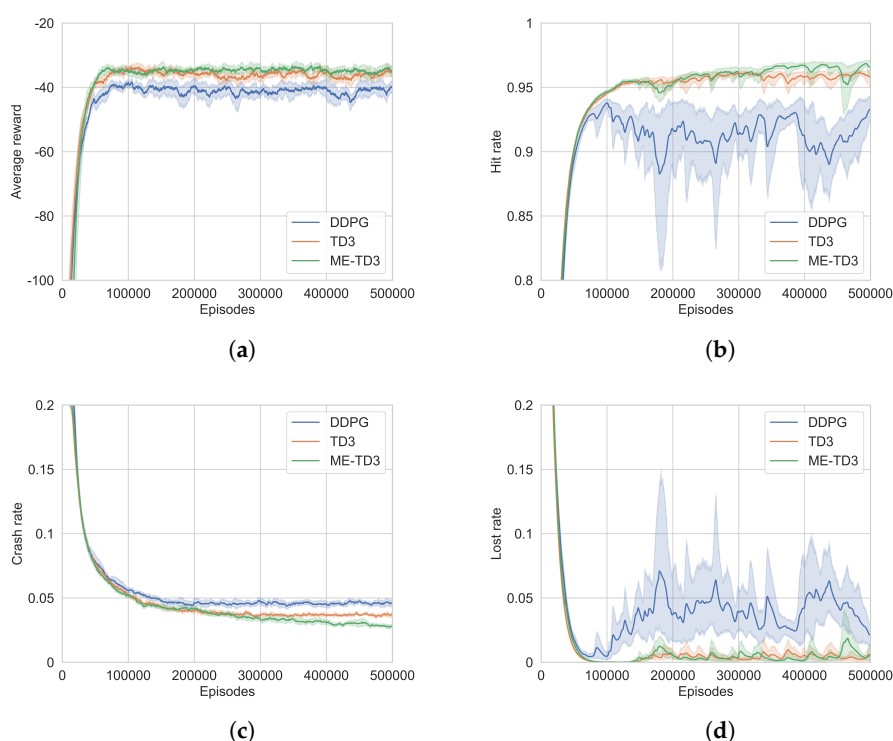

**Figure 8.** The convergence curves of algorithms. (**a**) Average reward. (**b**) Hit rate. (**c**) Crash rate. (**d**) Lost rate. Shadow areas represent the statistical distribution of data.

*6.3. Experiment of Algorithm Effectiveness*

To verify the effectiveness of the proposed algorithm, we constructed a dynamic simulation scenario of UCAV trajectory planning under radar detection threats. Specifically, the UCAV must start from the initial point $(0,0)$ km, cross a hostile radar surveillance area, and reach the target area centered around a circle at $(1000,1000)$ km. To directly demonstrate the coupling relationship between the radar detection probability and flight attitude, we intercepted the trajectories of UCAVs trained by different algorithms, indicating the number of steps below the image, as shown in Figure 9.

It can be clearly seen that the radar detection ranges vary at different times. When the UCAV's front faces the radar, its RCS decreases, leading to a reduction in the radar detection range. Conversely, when the UCAV's side faces the radar, its RCS increases, leading to an increase in the radar detection range. The UCAV must predict the changes in its own attitude that could result in alterations in the radar detection range. This proactive prediction is essential to prevent instances where sudden changes in the radar detection range could hinder its ability to rapidly evade threats. In addition, in some cases, the UCAV can briefly cross the radar detection range as long as it is not effectively tracked by the radars. DDPG, TD3, and ME-TD3 have reward values of $(-105.52, -102.82, -94.15)$

and $(144, 150, 137)$, respectively. ME-TD3 has the highest reward and the fewest running steps. DDPG has fewer running steps than TD3, but its reward is lower. This is because it adventures through radar detection areas to reach the target area as quickly as possible.

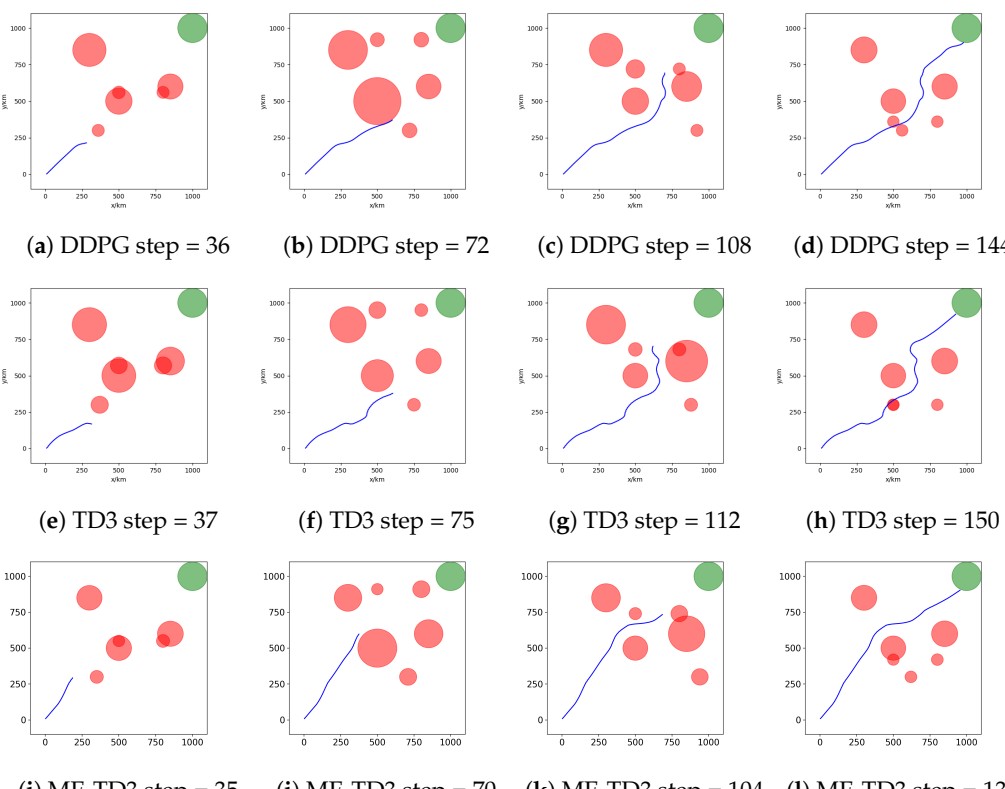

(**a**) DDPG step = 36    (**b**) DDPG step = 72    (**c**) DDPG step = 108    (**d**) DDPG step = 144

(**e**) TD3 step = 37    (**f**) TD3 step = 75    (**g**) TD3 step = 112    (**h**) TD3 step = 150

(**i**) ME-TD3 step = 35    (**j**) ME-TD3 step = 70    (**k**) ME-TD3 step = 104    (**l**) ME-TD3 step = 137

**Figure 9.** Flight trajectories of the UCAVs by trained different algorithms in a complex dynamic environment.

### 6.4. Experiment of Algorithm Adaptability

In this subsection, we constructed different simulation scenarios to test the adaptability of the proposed algorithm. Typical cases in which the UCAV successfully reaches the target area using the ME-TD3 algorithm are shown in Figure 10. We can see that the trained UCAV can adapt to different scenarios with an excellent trajectory planning performance.

We compared the performance of different algorithms in the above scenarios. At the end of training, these algorithms can ensure that the UCAV reaches the target area smoothly. In order to reduce the unreliability of the experimental results caused by fewer random seeds and test environments, we only analyzed the mean values of running steps and total rewards for different algorithms shown in Table 5. The total reward for ME-TD3 is the highest in all test scenarios, and the running steps are the fewest in case 1, case 2, and case 3.

**Table 5.** The results of algorithms in different scenarios.

| Algorithm | DDPG | | TD3 | | ME-TD3 | |
|---|---|---|---|---|---|---|
| | **Step** | **Reward** | **Step** | **Reward** | **Step** | **Reward** |
| Case1 | 144 | $-105.52$ | 150 | $-102.82$ | **137** | **$-94.15$** |
| Case2 | 142 | $-99.74$ | 141 | $-96.04$ | **136** | **$-93.44$** |
| Case3 | 151 | $-103.95$ | 143 | $-96.46$ | **141** | **$-94.72$** |
| Case4 | 142 | $-118.49$ | **140** | $-110.99$ | 141 | **$-106.15$** |
| Case5 | 152 | $-106.25$ | **146** | $-107.20$ | 154 | **$-103.68$** |

The best data are presented in bold.

In all tests designed according to the distribution of radar in reality, the UCAV controlled by the ME-TD3 algorithm can successfully complete the mission. However, there are still some crash cases in the late stages of training. We found that failure cases are caused by abnormal initialization of the UCAV and radar positions at the beginning of certain episodes. When the UCAV is initialized within the radar detection range, it cannot escape the threat area in a short time, resulting in a crash. Therefore, the random initialization limit of the training environment should be designed more reasonably to reduce useless training samples and improve the training efficiency.

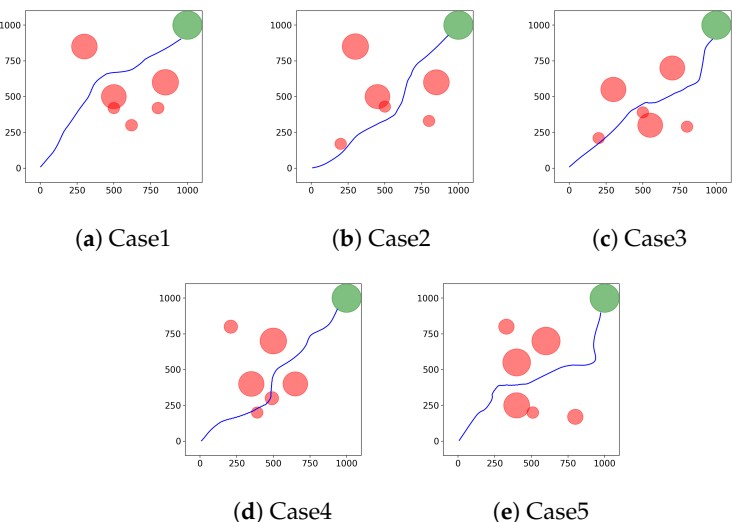

(**a**) Case1        (**b**) Case2        (**c**) Case3

(**d**) Case4        (**e**) Case5

**Figure 10.** Typical cases in which the UCAV successfully reaches the target area using the ME-TD3 algorithm.

## 7. Discussion

For a comprehensive evaluation of the proposed ME-TD3 algorithm, we conducted a large number of experiments by comparing it with the DDPG and TD3 algorithms to verify its performance, effectiveness, and adaptability. Different algorithms were trained and tested in randomly generated UCAV trajectory plans to avoid radar detection threat environments. These algorithms used the same hyperparameters, which were continuously tuned through repeated trials.

In the experiment investigating the algorithms' performance, we compared the hit rate, crash rate, lost rate, and average reward of different algorithms with environmental settings of 10 random seeds. As shown in Table 4, the ME-TD3 algorithm has the best success rate, crash rate, and average return. Furthermore, the best value of its lost rate is the lowest. Overall, the ME-TD3 algorithm performs better than the other algorithms in UCAV trajectory planning. The reason for this is that ME-TD3 employs an attention mechanism to process historical information, making it able to cope well with the dynamics and randomness of the environment.

In the experiment investigating the algorithms' effectiveness, we showed the flight trajectories of UCAVs trained by different algorithms in a typical scenario. As shown in Figure 9, it is obvious that the radar detection ranges change due to the effect of the spatial geometry relationship between the UCAV and the radar on the RCS. Both DDPG and TD3 only focus on the current state of the environment to make radical or cautious decisions. However, the UCAV trained by ME-TD3 has the predictive ability to adjust its flight attitude properly based on the dynamics of the environment. Furthermore, the flight trajectory generated by ME-TD3 has fewer running steps and higher rewards.

In the experiment investigating the algorithms' adaptability, different scenarios were established to test the performance of the algorithms, as shown in Figure 10. We compared the running steps and rewards of different algorithms while ensuring that the UCAV

successfully reaches the target area in each scenario, as shown in Table 5. The total reward of ME-TD3 is the highest in all test cases, which demonstrates it has better adaptability to dynamic and uncertain environments than DDPG and TD3.

Although we have provided preliminary statistics on the data generated by different random seeds in the performance experiments of the algorithm, further statistical significance testing is still important. Due to the small sample size, we are unable to effectively prove that the data satisfy normality and variance homogeneity. Therefore, we adopted the Friedman test method. Let $f_1$, $f_2$, and $f_3$ represent DDPG, TD3, and ME-TD3, respectively. The significance levels were calculated to assess whether the differences between $f_1$ and $f_3$; $f_2$ and $f_3$; and $f_1$, $f_2$, and $f_3$ are significant. The basic hypothesis is that the difference will not be significant. From the significance test results under different indicators shown in Table 6, the condition $p < 0.05$ occurs in tests of the hit rate, crash rate, and average reward, which means the hypothesis should be rejected. The significance between ME-TD3 and TD3 in the lost rate is 0.2059. The reason for this is that the loss rate has approached the optimization limit in the later stages of training, leading to an insignificant difference. In general, the differences in experimental results between DDPG, TD3, and ME-TD3 are still proven to be significant.

**Table 6.** Friedman test results.

|  | $p(f_1, f_3)$ | $p(f_2, f_3)$ | $p(f_1, f_2, f_3)$ |
|---|---|---|---|
| Hit rate | 0.0016 | 0.0114 | 0.0001 |
| Crash rate | 0.0016 | 0.0016 | $4.54 \times 10^{-5}$ |
| Lost rate | 0.0016 | 0.2059 | 0.0004 |
| Average reward | 0.0016 | 0.0016 | $4.54 \times 10^{-5}$ |

We have verified the effectiveness and adaptability of the proposed method through comprehensive experiments. By appropriately processing historical states, the UCAV can sensitively perceive the dynamics of the environment and thus exhibit a better trajectory planning performance. However, there are still some weaknesses and limitations for real-world development. In reality, the estimation of UCAV and radar states obtained by inertial navigation systems and specialized reconnaissance systems is inaccurate, which leads to a degradation in the UCAV trajectory planning performance. In addition, the lack of countermeasures against adversarial attacks on sensor data can lead to potential hazards in the UCAV control system.

We believe that the proposed algorithm has broad application prospects in practical applications. In the field of UCAVs entering battles with electronic countermeasures [41,42], this algorithm can be used to achieve more effective trajectory planning and improve UCAVs' safety under radar detection threats. However, the potential impact of this algorithm in practical applications needs to be noted. If the algorithm encounters malfunctions or errors in certain situations, this may have a negative impact on the security and stability of the system. Therefore, we need to ensure that the algorithm has sufficient robustness and fault tolerance.

## 8. Conclusions

In this paper, we proposed a method based on deep reinforcement learning to develop an intelligent UCAV that can perform automatic trajectory planning tasks under radar detection threats in dynamic and unknown environments. By analyzing the dynamics and randomness of the environment, we established a predictive control model and described it as a POMDP. A memory-enhanced TD3 algorithm based on an attention mechanism was proposed, which can utilize historical information to improve the performance of UCAV trajectory planning. Then, the UCAV was trained and tested in randomly generated simulation environments. The simulation results showed that the trained UCAV can safely and quickly penetrate a surveillance area composed of hostile ground-based radars and airborne radars and successfully reach the target area. Furthermore, compared with the DDPG and TD3 algorithms, the proposed algorithm has better performance and adaptability in complex environments.

In future work, the errors of state estimation and adversarial attacks on sensors will be considered to build a more realistic simulation environment for UCAV training. More methods to mitigate the impact of sensor data disturbances on algorithms will be studied to improve the robustness and stability of UCAV trajectory planning. In addition, considering their actual deployment, we will conduct in-depth research on potential issues in the migration process from simulation systems to hardware systems to meet the requirements of real-time performance and computational resources.

**Author Contributions:** Conceptualization, J.L. and T.Z.; methodology, J.L.; software, J.L.; validation, J.L. and K.L.; formal analysis, K.L.; investigation, J.L.; resources, T.Z.; data curation, J.L.; writing—original draft preparation, J.L.; writing—review and editing, K.L. and T.Z.; visualization, J.L.; supervision, T.Z.; project administration, T.Z.; funding acquisition, T.Z. All authors have read and agreed to the published version of the manuscript.

**Funding:** This research was funded in part by the National Natural Science Foundation of China under Grant 61971109, in part by the GF Science and Technology Special Innovation Zone Project, and in part by the Fundamental Research Funds of Central Universities under Grant ZYGX2020ZB031.

**Data Availability Statement:** Data are contained within the article.

**Conflicts of Interest:** The authors declare no conflict of interest.

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
