# Peer review of "Memory-Enhanced Twin Delayed Deep Deterministic Policy Gradient (ME-TD3)-Based Unmanned Combat Aerial Vehicle Trajectory Planning for Avoiding Radar Detection Threats in Dynamic and Unknown Environments"

_remotesensing, doi:10.3390/rs15235494_

Round 1
Reviewer 1 Report
Comments and Suggestions for Authors
Dear Authors,
The main concerns are given as follows:
1) There are grammatical and punctuation errors in the paper. The authors require a native speaker to proofread. The authors can use the professional version of the Grammarly system.
2) The introduction should be divided into two sections: 2. Introduction, 3. Related works.
3) The authors should review research studies by 2023 and these studies assigned in the Related Research work section. The research gap should be written at the end of this section and should describe the proposed method briefly.
4) The study’s strengths, weaknesses, and limitations should be written in a separate section after the discussion section.
5) The authors should compare their proposed method with the previous methods. They need to provide the comparison Table in the Discussion section.
6) The authors should write future works in the last section.
Comments on the Quality of English LanguageThere are grammatical and punctuation errors in the paper. The authors require a native speaker to proofread. The authors can use the professional version of the Grammarly system.
Author Response
Thank you very much for giving us the opportunity to revise. We conducted in-depth research and analyzed the questions and suggestions raised. We provided detailed explanations and explanations for each issue raised. In addition, we carefully revised the revision suggestions and gave specific revision details. We believe that your suggestion has made a great contribution to the level of the paper. Please see the attachment for specific responses.

Reviewer 2 Report
Comments and Suggestions for Authors
Dear Authors,
The article ME-TD3 based UCAV Trajectory Planning for Avoiding Radar Detection Threats in Dynamic and Unknown Environments is in the opinion of the reviewer an original work.
The Authors described in Introduction the aim and results of their research in the words: In this paper, we propose a novel UCAV trajectory planning method based on Deep Reinforcement Learning (DRL) technology A predictive control model is constructed to describe the dynamic characteristics of the UCAV trajectory planning problem in detail.to overcome the adverse impacts caused by the dynamics and randomness of environments. … Simulation results show that the proposed method can successfully train UCAVs to carry out trajectory planning tasks in dynamic and unknown environments.
Moreover, the Authors stated that: And the ME-TD3 algorithm outperforms other classical DRL algorithms in UCAV trajectory planning, exhibiting superior performance and adaptability.
The review focuses on the formal side of the article and does not engage in polemics and discussions on the considerations adopted by the Authors because it determines the originality of the issue in the article. Considering the proposed solutions I accept it with appreciation.
The following observation require to draw attention and response from the Authors:
1. Introduction. The use of four citation items [2-5] is, according to the reviewer, too extensive and forces the reader to review many cited items. I think two or three items are appropriate. I leave this matter to the Authors.
In the article, the Authors have used 37 items of references. Most references are contemporary items.
According to the reviewer, this article is very carefully prepared in terms of editing. In my review, I do not include more comments other than those presented in the review form.
Author Response

(The authors gave the same response as above.)

Reviewer 3 Report
Comments and Suggestions for Authors
This paper presents a robust methodology addressing a complex issue within the realm of autonomous systems, specifically Unmanned Combat Aerial Vehicles (UCAVs). However, to enhance the quality and impact of this research work, a few critical aspects could be improved or clarified:
- Literature Review and Positioning: While the paper addresses a novel problem, there is insufficient context concerning how this work extends or differs from existing literature. A more extensive literature review, positioning your contributions in the context of other DRL applications in trajectory planning, would strengthen the paper. Highlight specific deficiencies in current methods that your research is addressing.
- Technical Depth in Methodology: a. The explanation of the ME-TD3 algorithm could be more detailed, particularly for readers not intimately familiar with deep reinforcement learning. Discuss the inner workings of the attention mechanism within the context of your application.
b. Elaborate on why certain features were included in the state, action, and observation spaces and how they contribute to more effective learning. Are there any non-intuitive features, or did certain expected features prove unhelpful?
c. More information should be provided on the predictive control model. How is it formulated, and why is it suitable for this particular problem?
- Reward Function Design: While the reward function's construction is innovative, the rationale behind specific choices in its design needs more explanation. How do these choices impact the agent’s learning process? Discuss the balance between different components of the reward and the potential risks of unintended agent behavior due to these incentives.
- Simulation and Results Analysis: a. There could be more transparency in how the simulation environments were created, ensuring reproducibility. Are these environments based on real-world data, or were they generated randomly? What considerations were made to ensure they were representative of real-world scenarios UCAVs might encounter?
b. While the results indicate improvements over DDPG and TD3, it would be beneficial to include comparisons with more algorithms, especially those known for handling partial observability and memory requirements well.
c. A deeper dive into the failure cases of the ME-TD3 algorithm would be informative. Under what circumstances does it fail? What are its limitations?
d. Statistical significance: It would strengthen the paper to include a statistical analysis of the results. While the ME-TD3 algorithm appears to perform better, it's essential to demonstrate that these differences are statistically significant.
- Real-world Applicability and Ethical Considerations: The paper could benefit from a section discussing the real-world implications, potential applications, and ethical considerations of this technology. How do you foresee this algorithm being used in practice? What are the possible ramifications of its deployment in real-life scenarios?
- Future Work Limitations and Extensions: a. Discuss the limitations of your current work more candidly and how they might impact real-world deployment. It would add depth to your future work section.
b. Consider exploring not just errors in state estimation but also potential adversarial attacks on sensor data and potential countermeasures.
c. Future work could also discuss potential adaptations or necessary changes for deployment on actual hardware, considering real-time constraints and computational resources.
- Data and Code Availability: To increase the credibility and reproducibility of your findings, consider making your code, data, or simulation environment available for others to test and build upon.
Addressing these points will not only clarify the technical aspects but also increase the research's comprehensiveness and applicability, thereby enhancing its contribution to the field of autonomous systems and UCAV operation within contested environments.
Author Response

(The authors gave the same response as above.)
